# Temperature Gradients as a Data Storage Principle

**DOI:** 10.3390/e27020129

**Published:** 2025-01-26

**Authors:** Jeroen Schoenmaker, Pâmella Gonçalves Martins, Julio Carlos Teixeira

**Affiliations:** Centro de Engenharia, Modelagem e Ciências Sociais Aplicadas, Universidade Federal do ABC (UFABC), Santo André CEP 09210-580, SP, Brazil; pamella.martins@aluno.ufabc.edu.br (P.G.M.); juliocarlos.teixeira@ufabc.edu.br (J.C.T.)

**Keywords:** Landauer’s principle, hysteresis, data storage, information processing, temperature gradient

## Abstract

In this work, we analyze the thermodynamic principles underlying modern data storage systems, including Random Access Memory (RAM), hard disk drive (HDD), flash memory, magnetic RAM (MRAM), ferroelectric RAM (FeRAM), and phase-change RAM (PCRAM), as well as other less well-known data storage mechanisms. The analysis is conducted in the context of data storage and processing in relation to Landauer’s principle, with special emphasis on hysteresis. Analogous to how heat engines are characterized by thermodynamic cycles, data storage systems are examined in terms of the hysteresis loop of their fundamental data unit. We explore the role of heat in data storage systems. Afterward, we introduce the concept of temperature gradient memory (TeGraM) along with a detailed layout of a realizable device. Experimental results demonstrating this technology are also presented.

## 1. Introduction

Carnot’s seminal work, *Réflexions sur la Puissance Motrice du Feu*, makes two primary contributions: it introduces the analysis of heat engines in terms of cyclic processes and highlights the importance of the temperature difference between the heat source and the environment [1]. His work provided a rudimentary understanding of the Second Law of Thermodynamics and paved the way for describing each type of heat engine according to its unique thermodynamic cycle, with the “Carnot cycle” serving as a model for an ideal engine. In a similar vein, our work analyzes established data storage technologies through the lens of a characteristic cycle, the “hysteresis loop”, associated with the memory unit cell. We contend that, while each memory technology exhibits some form of hysteretic behavior, this characteristic has not been widely recognized as a fundamental feature. Instead, each technology has approached the concept in its own terms. For example, the term “hysteresis loop” is seldom used consistently across the field; in flash memory, it is often referred to as the “memory window”.

An in-depth analysis of memory technology from the perspective of hysteresis can offer new insights into data storage devices. We illustrate this through the following steps:Recovery of the fundamentals of irreversibility in computing processes established by Landauer;Assertion of the forefront role of hysteresis loop characterization for memory technology;Examination of other hysteretic phenomena that are suitable for memory technology;Proposal of TeGraM, a new type of memory technology.

Firstly, it is important to discuss the meaning of “information” and “data”. Depending on the context, the term “information” can be related to notions such as understanding, knowledge, meaning, communication, data, memory, representation, energy, and more. In this manuscript, we focus on the notion of information used in computer and communication technology. Moreover, although in many contexts, one can discuss the differences between “data” and “information”, for the objectives of this manuscript, these terms are generally interchangeable. The discussion here is centered on computation performed by digital data processors associated with data storage devices.

In a publication in 1961 entitled “Irreversibility and Heat Generation in the Computing Process” [2], Landauer put forward the case that “information processing is inevitably accompanied by a certain minimum amount of heat generation”. This principle, known as the Landauer limit, is now also referred to as the Carnot–Landauer limit in the context of quantum systems [3]. This assertion is based on previous work by Brillouin and earlier authors in which it is stated that every measurement process requires a dissipation of the order of “kT”, where k is Boltzmann’s constant and T is the temperature. As Landauer was working for IBM and published his work in the company’s journal, by “information processing”, he meant computation. Interestingly enough, Landauer pointed out that one important limitation of Brillouin’s work was the poor definition of measurement in a similar fashion as we here debate the poor definition of “information” and “information processing” in a broader sense. The discussion is akin to the debate in quantum mechanics regarding the separation between the system and observer when he asks: “When is a system A coupled to a system B performing a measurement?”

In his manuscript, Landauer separates the cases where information is held dissipatedly (what is nowadays related to the volatile and most common form of Random Access Memory in computer architecture) and the cases where information can be stored without dissipating energy (nowadays, more associated with the hard disk drive). Landauer does not elaborate much in the first case, as he argues that volatile memory is inherently dissipative. On the other hand, special attention is given to the nonvolatile type of information storage, which is, in turn, divided into two basic types depending on the shape of the potential well responsible for maintaining the physical state of the stored information: the bistable potential well and the potential well with 0 and 1 states not separated by a barrier (Figure 1). The first type is associated with ferromagnetic and ferroelectric systems with the condition that the switch mechanism happens without domain wall motion; thus, hard disk drives could be a modern representative of such technology. When discussing the second type, Landauer associates mainly with an old kind of memory device called a cryotron but also with more old-fashioned and analogical kinds of information storage: writing, punched cards, microgroove recording, etc.

The computational process of “restore to one” is central to Landauer’s reasoning and deals with the transitions between physical states, defined as 0 and 1. It is worth noticing that a consistent characterization of the number of logic cycles “write” and “erase” (equivalent to restore to one in Landauer terminology, where the term “restore to one” is equivalent to “erase”) is key for the establishment of any data technology. By making a fundamental analysis of such an information-switching mechanism, he arrives at the conclusion that for every computational process, a minimum of kTln2 energy is dissipated in the environment in the form of heat. Interestingly, right after this conclusion, Landauer states “and our method of reasoning gives no guarantee that this minimum is in fact achievable”. Note that even nowadays, information technology operates with switching energy dissipation 3 orders of magnitude above such minimum [4].

In the same manner, Landauer does not compromise by stating that the minimal switching energy is physically achievable, and he is also not very assertive when he states “We believe that devices exhibiting logical irreversibility are essential to computing. Logical irreversibility, we believe, in turn, implies physical irreversibility, and the latter is accompanied by dissipative effects”. Although the discussion regarding the relationship between data computing and energy loss has been the subject of abundant literature, such discussions have been focused on fundamental issues (mostly centered on rather idealized systems) [4,5], and main assertions have been a matter of profound debate until recently [6].

In this manuscript, we focus on the fundamental interpretation of the second law that can be derived from the early works of Carnot, in which irreversible processes are strongly related to energy dissipation and heat released to the environment [7]. Moreover, we focus our analysis on the hysteretic manifestation of irreversibility in the form of the laboratory testing of the memory unit cell (or medium). As a matter of fact, Landauer mentions hysteresis only once in his manuscript when he analyses the restore-to-one process in a bistable potential well and comes to the conclusion of an unavoidable dissipation of energy. Then, he states “This energy is dissipated and corresponds to the one-half hysteresis loop area energy loss generally associated with switching”. Thus, our intention is to deepen the analysis of the inherent correlation between energy dissipation, the hysteretic behavior of physical systems, and data storage. We not only settle that the feasibility of data storage technologies is fundamentally related to their corresponding hysteresis loops, but we go further and use this connection to propose a new data storage principle strongly associated with heat engines: temperature gradient.

## 2. The Connection Between Data Storage, Hysteresis, and Energy Loss

Scientific studies using hysteresis curves were established for the investigation of magnetic materials in the late XIX century. Afterward, the approach was adapted to other systems, such as ferroelectricity and the mechanical properties of materials, moving to other fields such as medicine and even social sciences. The term derives from a Greek word meaning “lagging behind” that already hints something of a “memory”.

There is a well-recognizable quotidian experience of a hysteretic behavior that happens when one attaches a magnet to the tip of a screwdriver. After the removal of the magnet, the tip of the screwdriver behaves as a magnet itself, as if the magnetization provided by the magnet lags behind and the screwdriver “remembers” its magnetized state for a rather long time after the removal of the external stimulus. The magnetization of a screwdriver (or any sample that behaves ferromagnetically), as well as its hysteresis loop, can be represented as shown in Figure 2, where the magnetization ($\vec{M}$) of the sample as a function of the applied magnetic field ($\vec{H}$) is depicted. At the beginning of the process, the sample is usually in a demagnetized state and no magnetic field is applied (H=0,M=0). As the magnetic field is applied in one direction (say, we approach the north pole of a strong magnet), the sample is increasingly magnetized until the field is high enough to saturate the sample. As evident in Figure 2a, the magnetization process is inherently irreversible. Once the sample is magnetized, if we decrease the applied field back to zero, the magnetization does not follow the same path taken by the initial process. Instead, the sample retains a remnant magnetization ($M_{r}$) after the applied field vanishes. This explains why the screwdriver remains magnetized and becomes a magnet itself after the magnet is removed. Figure 2b shows the behavior of the magnetization in the case where we apply an increasingly stronger magnetic field in the opposite direction (say, we turn the magnet in the opposite direction and approach the sample with the south pole) and we saturate the sample in the other direction. The hysteresis loop is closed when we magnetize and saturate the sample again in the initial direction, as show in Figure 2c. Another important parameter of the hysteresis curve is the coercive field ($H_{c}$). The larger the $H_{c},$ the more difficult it is to reverse the magnetization of the sample. Note that here, we describe the curve that passes through saturated states in both directions, which is the most common procedure. However, hysteresis curves can be measured without reaching the magnetic saturation of the sample, and they are called “internal hysteresis curves”. The behavior of the magnetization of such internal curves is notoriously difficult to model and predict, and it is strongly dependent on the magnetization history of the sample, highlighting again the thermodynamic irreversible nature of the magnetization process of a ferromagnetic sample.

Two important things are worth mentioning. First, we have to keep in mind that the hard disk drive (HDD) is arguably the most well-established technology for data storage, even to this day, and it is based on the ferromagnetic nature of its media. Ideally, the media should have a large $M_{r},$ rendering that each datum can be easily detected by the read head. Moreover, the media should have a large $H_{c}$ as this value is proportional to the thermodynamic stability of the stored data. Thus, the larger the area inside the hysteresis curve, the better it is for data storage [8].

The second aspect is that the area inside the hysteresis curve is proportional to the energy dissipated in the magnetization reversal process. In the particular case of magnetic hysteresis, this is not readily intuitive, but it is well-known to power electrical engineers. Ordinarily, electric devices, such as motors, generators, and transformers, have magnetic cores that have their magnetization reversed in each cycle. Such electric ferromagnetic materials, in opposition to the magnetic media for data storage, are engineered to have a small internal area inside hysteresis loops in order to avoid energy losses. This highlights an interesting dichotomy in the application of magnetic materials: when they are applied to data storage, high hysteresis is desirable, whereas when they are meant for electromechanical conversion, hysteresis means inefficiency.

If not aligned in a specific way, a magnetic field exerts torque on a magnetic dipole. Thus, the magnetization reversal is inherently related to the work performed by the applied field in reversing the magnetization of a sample that can be expressed by the formula [9](1)$$\frac{Work}{Volume}=\int_{0}^{M_{final}}HdM$$

If the calculation is performed for the magnetization throughout the hysteresis, one has to account for the work completed and the work given back by the system (Figure 3). As a result, it is verifiable that the internal area of the hysteresis loop is the energy dissipated per volume of material.

As the processes of write/erase in hard disks are inherently related to magnetization reversal and, consequently, energy dissipation, we see a corroboration of Landauer’s principle. Thus, we settled the issue of the relationship between hysteresis, data storage, and energy dissipation for hard disks. In the next sections, we aim to generalize this analysis and push the discussion further with a fresh point of view.

## 3. Hysteresis Loops in Data Storage Technologies

Data storage systems can be divided into volatile (mainly RAM) and nonvolatile memories. The first kind was described as evidently lossy by Landauer and did not receive dedicated attention in his works. However, for the sake of our analysis, the fundamentals of RAM associated with hysteretic behavior are discussed here. On the other hand, the current conventional nonvolatile data storage systems that have an established market can be categorized as hard disk drives (HDDs), flash memories, and optical discs [10]. Although the latter has been upgraded through three generations, starting from CDs to blue-ray discs, this type of technology is losing market share due to optical limitations for data density increase. Over the last 5 decades, HDDs have been the mainstream storage device mainly due to their high capacity, fast data rate, and low cost, which are still dominant in cloud data centers. However, recently, although its low cost per stored datum still favors the HDD format in data centers, flash memory technology is gaining most of the marked share because of its high capacity, short data access time, portability, and low expense. Flash has become the dominant data storage device for mobile electronics [11,12]. Even enterprise-scale computing systems and cloud data storage systems are using flash to complement the storage capabilities of HDDs [13].

Flash memory relies on a capacitive system to store information, and it is the term given to a broad set of nonvolatile memories with different brand names and architectures, such as Pen Drives, USB flash drives, SD cards, microSD cards, NOR flash, NAND flash, vertical NAND flash, etc.

Besides the already mentioned technologies, we identify other less widespread or even emergent nonvolatile memory technologies, including ferroelectric RAM (FeRAM), phase-change memory (PCM), magnetic RAM (MRAM), spin-transfer torque MRAM (STT-MRAM), and resistive RAM (RRAM) [10,11,12,13,14].

### 3.1. Volatile Memory Systems

In computer architecture, the usage of volatile memory is necessary in order to improve computation performance. As recognized by Landauer, this type of memory is inherently lossy, and the information is lost at the time the computer is turned off. However, in a deeper analysis of the relation between memory and thermodynamics, we can also recognize the role of hysteresis loops.

There are two types of volatile memory: Static RAM (SRAM) and Dynamic RAM (DRAM). In the case of SRAM, the unit of data storage (or bit cell) typically consists of up to six transistors, forming two CMOS inverters connected in a positive feedback loop to create a bistable storage element. Regardless of the stored state, the information is maintained by a constant voltage supplied to the circuit. Once this voltage is removed, the data stored in SRAM are lost. Furthermore, the read and write processes are controlled by electronic switches called “set–reset flip-flop circuits”, which are associated with a bistable potential, as described by Landauer (Figure 1a). Flip-flop circuits function similarly to Schmitt triggers and can be considered as one-bit quantizers. These circuits play a role in converting single analog signals (such as a one-bit signal in SRAM or key presses in keyboards for Schmitt triggers) into digital bits for computer processing. A fundamental role of these circuits is to distinguish signal from noise. This is achieved through a dual-threshold mechanism, which creates an energy barrier known as “trigger hysteresis,” allowing the circuits to act as bistable multivibrators.

The usage of SRAM is mostly restricted to cache memory, and DRAM is the dominant form of volatile memory. In DRAM, each storage unit consists of a capacitor associated with a transistor (Figure 4a,b). The binary state is defined as the charged and uncharged condition of the capacitor. However, it is a fact that the capacitor’s electric charge constantly leaks through the transistor, leading to the necessity of charge reload. Thus, as opposed to SRAM, which requires a steady voltage supplied to the memory cell (hence, the name “static”), DRAM requires a reset cycle to take place, lasting about a few tens of milliseconds (hence, the name “dynamic”). Furthermore, this means that the working principle of the bit cell is based on a potential well without an energy barrier, as described in Figure 1b, and according to Landauer, “information is preserved because random motion is slow”. Figure 4b shows a sketch of a trench capacitor DRAM cell.

At the fundamental level, the role of the data storer is performed by the capacitor. A truly isolated capacitor would remain charged indefinitely, thus realizing the role of a nonvolatile memory unit cell associated with the transistor. However, due to the charge leakage, the bit cell ends up working fundamentally as a resistive–capacitive circuit, which, under charge cycling conditions, portrays a hysteretic behavior, as illustrated in Figure 4c. As we demonstrated in Section 2 for the magnetic case, the area inside the loop described in each cycle of the RC circuit in the voltage–current chart can be understood as the energy dissipated in each cycle, as we recall the definition of electric power is voltage times current.

### 3.2. Hard Disk Drives and Other Magnetic-Based Memories

In the context of Figure 2, we discussed the physical irreversibility of the magnetization process of a magnetic body. The most common material for HDD media is a granular thin film of a ferromagnetic cobalt–chromium–platinum alloy. The stored binary information is established by a sequence of regions of the disk magnetized in opposite directions (or opposite polarities), as shown in Figure 5.

However, the HDD is not the only way to use the magnetization irreversibility for information storage. In fact, the history of data storage has several examples of other strategies based on magnetic materials for that purpose, although old technologies were mostly analogical. Nowadays a few emergent data storage technologies also rely on magnetic materials.

Magnetoresistive Random Access Memory, or just MRAM, is based on memory cells that have a Magnetic Tunnel Junction (MTJ). Each MTJ works with two magnetic layers: one with a fixed polarity and the other with an adaptable polarity (Figure 6). The hysteretic behavior of the element with adaptive polarity settles the information stored. These two elements are separated by a thin insulating material through which the electric current trespasses by tunneling [10,15]. The information stored can be read resistively as the spin-dependent scattering of the electrons renders different resistivities of the cell depending on the alignment or counter alignment of the adaptive polarity element.

The second emerging technology that uses remnant magnetization as physical irreversibility is the STT-MRAM (spin-transfer torque Magnetoresistive Random Access Memory). It is similar to MRAM, but it uses polarized spin currents. The storage process is given through the electric current that is polarized by the alignment of the direction of the electron spin flowing on an MTJ. This current of polarized spins changes the orientation of the magnetic field of the storage layer of the MTJ.

Because of the fact that they are magnetic, the MRAM and the STT-MRAM have a hysteresis curve similar to the HDD, as shown in Figure 6b [16]. As these technologies rely on a ferromagnetic element to establish the write/erase states, it is redundant to discuss the association of the internal area of the curves with energy dissipation.

### 3.3. Flash Memories

Hard disk drives and flash memories are the two main contemporary nonvolatile memory technologies. However, while the usage of HDDs is becoming restricted to data centers, flash memory devices are increasingly gaining the market.

A detailed description of the workings of such devices is not required here. Semiconductor devices, from memory circuits to camera photodetectors, take advantage of decades of accumulated knowledge, infrastructure, and processing techniques developed for integrated circuit microprocessors. In this way, as shown in Figure 6, each cell of such devices is a “transistor-like” unit, i.e., they have basically the same structure of a transistor, however, with some additions or modifications of the existing elements. So, in this manuscript, we explain how the addictions of these elements in such “modified transistors” provide nonvolatile memory properties.

In flash memory, the information is stored as a matrix of transistors referred to as “cells”. Traditionally, each cell stores one bit of information. There are two categories, i.e., NOR and NAND, and they differ in architecture but rely on the same physical mechanism in order to store information [17,18]. Figure 7a shows the programming of flash memory using the Fowler–Nordheim mechanism.

According to this mechanism, a high voltage is applied between the Control Gate and the Drain, making it possible for some electrons to cross the oxide layer and reach the Floating Gate, executing the process of writing. A low voltage can be applied to revert the process and erase the information. Thus, the binary manifestation of flash memory is realized by the presence or absence of trapped electrons in each cell as they are manipulated by electric fields, constituting a capacitive type of technology.

Interestingly, crucial developments in flash technology depend on the charge-trapping mechanism in the floating gate. Traditionally, the floating gate has been constituted by doped polycrystalline silicon. New developments replaced the floating gate with a charge trapping layer (CTL) made of different oxides with high dielectric constants, with most vendors adopting silicon nitride [18,19].

The characterization of the trapping mechanism quality is established by the “memory window”, where the hysteretic behavior of the capacitance as a function of the Control Gate voltage is measured (see Figure 7b). The memory window is given in volts and, in a similar way as the magnetic case, the breath of the curve is proportional to the stability of the stored information.

It is also readily verifiable that the internal area under the memory window hysteresis is proportional to the dissipated energy in the process once we keep in mind that the energy stored in a capacitor is given by(2)$$E=\frac{1}{2}CV^{2}$$

### 3.4. Ferroelectric Memories

Ferroelectric memories were proposed in the early fifties and still, nowadays, are referred to as a possible technology to stand side by side with HDDs and flash memory as a widespread technology [20]. Ferroelectric Random Access Memory (FeRAM) is the most established form of ferroelectric memory. Some commercial devices have been put into the market by companies such as Panasonic, Toshiba, and Siemens [21]. At the moment, FeRAM technologies, which are based on perovskite and fluorite ferroelectrics, suffer from issues including poor complementary metal oxide semiconductor (CMOS) compatibility and limited scalability, adding to the manufacturing cost. Thus, FeRAM is still restricted to a small niche of the market [14,22].

Possible structures of FeRAM cells are akin to the so-called “one-transistor (1T)” or “one-transistor–one-capacitor (1T-1C)” configurations of DRAM. Figure 8a depicts a schematic structure of a FeRAM cell in a 1T structure, in this case, identified as a Metal–Ferroelectric–Insulator–Semiconductor Field Effect Transistor (MFIS-FET). The multilayer composition is given by TiN/Si:HfO_2_/SiO_2_/Si. In Figure 8b, we observe two distinct hysteresis curves obtained for a specific FeRAM cell. A 0.8 V memory window is distinguishable in the figure as the result of the ferroelectric behavior of the Si-doped HfO_2_ layer plotted in the upper inset. As already shown in the previous section, the memory window is proportional to the thermodynamic data stability. Interestingly, the internal area of the ferroelectric hysteresis curve is also proportional to the dissipated energy once we consider that the applied voltage is proportional to the electric field and the electric polarizations are related to charge displacement.

### 3.5. Phase-Change Memories

In the introduction of this section, it was mentioned that nonvolatile memory in the form of CDs is losing market share and that the format has been through several stages of development, going from regular CDs to DVDs and Blue Rays. It is worth mentioning that the early CDs and DVDs were non-rewritable, meaning that once they were recorded, they were read-only media. From the point of view of this manuscript, this is fundamentally akin to writing on a sheet of paper using a pen. However, during the 1990s, CDs were made available on the market in a rewritable format. This was possible due to the discovery of a new class of materials, called chalcogenides, which have a considerable change in light reflectivity when changing from the crystalline phase to the amorphous phase, and this phase change can be consistently performed with a laser heating thermal process. This led to a spark of interest in producing solid-state nonvolatile memory technologies, taking into account that this same class of materials also exhibits a large change in electrical resistivity when changing phases, and such transformation could be induced by electrical currents [17,24].

The cell structure of phase-change Random Access Memory (PCRAM) is similar to an MRAM cell (Figure 6), as in both cases, the contrast between the “0” and “1” states is accessed by a difference in the cell’s electric resistivity. Figure 9b depicts the hysteretic behavior of the current flowing through the PCRAM cell as a function of the applied voltage as the programmable volume of the chalcogenide layer changes from a crystalline state to an amorphous state. Note that the curve relates current as a function of voltage. Thus, its internal area is proportional to dissipated energy, similar to what happens in Figure 8. Furthermore, an important parameter while addressing the quality of the memory cell is the voltage threshold (V_th_), which is associated with the stability of the “off” state and readability of the stored data. The hysteretic behavior of blade-type PCRAM has also been obtained by simulations [25].

It is worth noticing that failures of such devices, mainly related to processes associated with thermal/mechanical fatigue such as void formation and elemental segregation, are among the reasons hindering the widespread usage of this technology [24].

### 3.6. Final Remarks on Memory Technologies

As discussed in the previous section, hysteresis loops are inherently associated with irreversible processes and energy dissipation. However, it is worth mentioning that heat management can vary drastically between each memory technology. Aside from the intrinsic characteristics of the write/erase process in each case, there are other factors to consider, such as whether the memory is volatile or nonvolatile, for instance. Consider the case of HDDs. Although the magnetic media is designed to maximize its internal hysteresis loop area, as a nonvolatile type of memory, the write/erase processes are performed intermittently, requiring minimal thermal management. In contrast, DRAM must refresh stored data every few tens of milliseconds, resulting in significant power loss and necessitating adequate thermal management. High-performance DRAM, in fact, is known to incorporate heat sinks to address these thermal challenges.

Beyond the technologies already mentioned in previous subsections, there are many other concepts and designs for nonvolatile memories that are less representative in terms of the stage of development or marked share. The effort to exhaust all possible technologies would not add much to the aim of this manuscript. For instance, we could address resistive Random Access Memory (RRAM) and racetrack memory [13,17].

Racetrack memory is an emergent type of magnetic memory in which bits of data are stored in a U-shaped magnetic nanowire, whose magnetization state is dependent mostly on magnetic domain wall nucleation and movement [26].

The case of RRAM is tricky because the term refers to memories based on a change in electric resistivity to encode the binary state of each cell. In this sense, MRAM and PCRAM can be considered types of RRAM. However, there are others [17]. One proposal is to use diffusion mechanisms of O_2_- to control the formation and rupture of conduction filaments in binary metal oxides such as NiO, TiO_2_, Nb_2_O_5_, and ZrO_2_. Another possibility is the use of perovskites as key elements to attain resistivity variation. Perovskites are known to have interesting properties such as colossal magnetoresistance, ferroelectricity, and superconductivity. We could still go further with even more proposed mechanisms to form and rupture conduction filaments inside RRAM memory cells, but they are overall based on physical/chemical processes such as magnetization, ferroelectricity, crystalline structure changes, and elemental diffusion. It is not difficult to recognize, especially after our analysis of HDDs, flash, MRAM, FeRAM, and PCRAM, that all these processes involve a hysteretic behavior.

Finally, and importantly, we found that accurately characterizing the hysteretic properties of fundamental memory device elements is important for determining their true capabilities.

## 4. Hysteresis Loops and Thermodynamics Cycles

Carnot was a pioneer in understanding heat engines through the concept of thermodynamic cycles. Historically, thermodynamic cycles were introduced as a theoretical framework to model and comprehend the operation of heat engines. In most cases found in the literature, heat engines are explained in terms of their associated ideal (or theoretical) thermodynamic cycles, which are then related to real engines. Real (or measured) thermodynamic cycles are typically used in more specific cases for practical applications.

In contrast, the study of magnetism through hysteresis loops was established by Alfred Ewing’s laboratory measurements in the late 19th century. Models aimed at explaining hysteresis cycles, such as the Preisach and Stoner–Wohlfarth models, emerged decades later. Even today, hysteresis loops are primarily regarded as a laboratory characterization technique rather than a theoretical framework. They have become a cornerstone in the characterization and development of magnetic materials.

Although it is possible to find works analyzing data storage technologies featuring curves displaying the hysteretic behavior of a key element of their data cell, the relevance of these curves has been secondary in the literature. One piece of evidence of this fact is that in most cases, these curves are referred to by different terms such as “memory window” or simply by “I-V curve”, indicating a narrower appreciation of this kind of analysis. Another piece of evidence is that the correlated variables in the curves presented are not consistent in the form that could enable a systematic widespread analysis of memory devices. Every mechanical engineering book will explain heat engines through their thermodynamic cycles, yet books on digital memory devices seldom give hysteresis loops the same status.

The kinship between hysteresis loops and thermodynamic cycles is multifaceted: their internal area and shape give important information regarding the performance of the device they are associated with and are related to the energy dissipated or produced.

In the case of digital memories, hysteresis loops indicate the stability of the information stored (how long the data can be stored before heat-related processes or degradation ruin the desired state) and how readable the stored information is. This is crucial in an era of miniaturization.

One particular data storage technology that has advanced significantly through the analysis of hysteresis loops is the HDD. Figure 10 shows several decades of progress in the coercive field and magnetic flux parameters of the magnetic thin films used in this technology. As areal data density increased, the coercive field needed to be raised to maintain data stability. However, this advancement led to a reduction in the magnetic flux of the media, making stored data progressively harder to read. Fortunately, the development of new read-head technology based on Giant Magnetoresistance (GMR) enabled further increases in HDD areal density by several orders of magnitude. In recognition of this breakthrough, the 2007 Nobel Prize in Physics was awarded for the discovery of GMR, which made it possible to detect minute magnetic signals in modern HDDs.

## 5. Implementation of Temperature Gradient Memories

By establishing a correlation between memory and hysteresis, new concepts for memory storage devices can now be envisioned. In this manuscript, we present one such possibility. When discussing temperature gradient memory, it is intriguing to highlight certain similarities between this memory concept and heat engines, both in terms of perspectives and underlying principles:Analysis through cycles: We addressed the transition of each bit of stored data between states 0 and 1 during the write/erase process as closed cycles (hysteresis loops), analogous to the thermodynamic cycles associated with heat engines.The internal area of the cycle: In thermodynamic cycles, a larger internal area corresponds to higher engine efficiency. Similarly, in memory storage, a larger hysteresis loop area indicates better device performance, including greater data stability and readability.Exploration of temperature gradients: We envisioned the role of temperature gradients as the physical principle of this memory concept.

Intuitively, using a temperature gradient as an information storage principle is reasonable—consider how a hot coffee or a cold beer left on a table signifies that someone has been in the room recently. Temperature gradient processes are inherently hysteretic. However, this specific application of temperature gradients remains largely unexplored. There are, of course, some related studies that link heat, memory, and computing. For example, thermoelectric effects have been investigated to support PCRAM [27,28,29]. However, our proposal does not involve phase-change materials. On a different front, research on “phononic circuits” has introduced basic components like diodes and memory elements based on heat flow [30,31,32]. Although these phononic concepts have drawn significant academic interest, they are still experimental and mostly limited to quantum computing, thermal management, and signal processing applications. In contrast, our proposal is aimed at direct integration with electronic circuits.

As a relatively unexplored area, there are numerous avenues open for experimentation. Notably, moving beyond traditional hysteresis loop analysis has enabled us to take a distinct approach, leading to a concept of thermal memory from a perspective different from that of the phononic circuit field, where thermal memory is often defined as “a temperature maintained somewhere” [31,32]. As a historic parallel, before Carnot’s contributions, advancements in heat engines primarily focused on heat sources and the hot parts of the engine. Carnot demonstrated the need for a broader perspective, incorporating the temperature gradient relative to the environment and the complete thermodynamic cycle of the engine. Similarly, we recognize that a thermal memory should be based on a temperature gradient, and the feasibility of such a device should be assessed through a hysteresis loop. Accordingly, our approach naturally incorporates thermoelectric effects to enable temperature gradient memory compatible with electronic circuitry.

In this manuscript, we describe the implementation of temperature gradient memory (TeGraM) in two different kinds of devices:Devices that actively produce the temperature gradient: aTeGraM;Devices that passively take advantage of temperature gradients from the environment: pTeGraM.

We also demonstrate the read/write processes of this technology.

In order to put forward the TeGraM technology, we take advantage of thermoelectric effects. For aTeGraM, it is sensible to use the Peltier effect [33](3)$$\dot{Q}=\left(\Pi_{n}-\Pi_{p}\right)I$$
where $\dot{Q}$ is the heat flux generated, $\Pi_{n}$ and $\Pi_{p}$ are the Peltier coefficients for the n- and p-type semiconductors of the Peltier unit, and I is the current, in order to produce temperature gradients.

Analogously to DRAM, where a transistor is associated with a capacitor to form a unit memory cell, we can associate one Peltier unit to a transistor to form an aTeGraM unit cell, as depicted in Figure 11. Two different proposals for the binary states can be used: the presence or absence of a temperature gradient or the two polarities of the hot/cold orientation. The implementation of each proposal will depend on the type of transistor used and the kind of processing to manufacture the thermal storage system. The read and write processes are similar to the ones performed in DRAM technology. To write a bit, a voltage is applied to the wordline, allowing the bitline to submit the Peltier unit to a current and generating a heat flow between the masses. To read the stored bit, the voltage in the Peltier unit can be read via the Seebeck effect(4)$$V_{CH}=\int_{T_{c}}^{T_{H}}\left(S_{p}-S_{n}\right)dT$$
where $V_{CH}$ is the voltage generated between the cold (at temperature $T_{C}$) and hot (at temperature $T_{H})$ masses of the Peltier unit and $S_{p}$ and $S_{n}$ are the Seebeck coefficients of the n- and p-type semiconductors of the Peltier unit. The second Thomson relation(5)Π=TS
establishes that the Seebeck voltage is fundamentally related to the information written via the Peltier heat flow that established the temperature gradient.

Another possible implementation is pTeGraM, which utilizes a temperature gradient present in the environment. Examples of environments with temperature gradients that could be explored include ocean depth profiles, geothermally active areas, or objects in space with respect to their orientation to the Sun. Interestingly, another application could relate to computer architecture, where certain components, like processors, generate significant amounts of heat. pTeGraM could be integrated within the heat exchangers that manage thermal control in these devices.

In this case, as the orientation of the temperature gradient is fixed, the writing process is performed by a dedicated flip-flop circuit. Figure 12 shows one possible implementation via a transruptor [34]. Other flip-flop circuits may be considered. pTeGraM works similarly to SRAM; however, in this case, the voltage necessary to maintain the state of the recorded information is provided by the environmental temperature gradient. As the thermal gradient is persistent, the memory can be considered as nonvolatile.

As a proof of concept, we demonstrate an aTeGraM read/write process of one unit cell formed by a 40 × 40 mm^2^ Bi_2_Te_3_ Peltier TEC1-12706 module thermally connected to size-matching steel cubes positioned on each side of the module. A power supply and an oscilloscope were used to apply and monitor the voltage on the Peltier module terminals. Figure 13a depicts the results of a write-and-read-like process. An increasingly higher voltage was applied to the module for 2 s, starting from 0.25 V up to 1.25 V. After each step, the power supply was reset to zero, and the Seebeck voltage on the terminals resulting from the produced temperature gradient was registered by the oscilloscope. In this regime, the read voltage was directly proportional to the write voltage with a ratio of about 0.15. Figure 13b depicts the hysteresis loop obtained for the data cell. The regime of data acquisition used was kept the same as that of Figure 13a; however, the write voltage was completed twice the swept from 0 to 1.25 V, back and forth. Note that the curve starts from the unheated state (origin) and proceeds to the maximum temperature gradient (attested by the maximum Seebeck read voltage of the graph). This initial curve usually stands out with a distinct behavior from the curve established in the cycling regime, as we observe in Figure 2a and Figure 4c. Note that the second and third time the write voltage reaches zero, the values of the read voltage are practically the same, indicating that the cycle reached a regime mode. The memory window is evidenced by the shaded area.

In current thermoelectric technology, Peltier and Seebeck effects are used in distinct settings, i.e., Peltier devices are developed for cooling/heating purposes, while Seebeck devices are developed for temperature sensing. In this way, thermoelectric materials designed for TeGraM are a subject for future research that should aim for materials with integrated Peltier/Seebeck performance compatible with the electronic and semiconductor industry. However, this venue of research can take advantage of the research on thermoelectric-assisted PCMs and semiconductor-based thermoelectric materials [35,36].

Aside from research into dedicated thermoelectric materials for temperature gradient-based memory systems, there are several avenues for further development. In the case of aTeGraM, the distinct regimes of heat and electronic flow could be leveraged as a potential advantage. For high-density data systems, the logical progression would involve miniaturization and increased speed. Parameters such as mass and contact surface area play a critical role in heat flow engineering. As volumes decrease, the surface-to-volume ratio increases, altering the timescales and other key aspects of the process. The speed of computation is another fundamental parameter to be considered [37]. The promising news is that current volatile memory technology already operates on the millisecond timescale.

For pTeGraM, the solution will largely depend on the specific temperature gradient environment selected for the device, such as geothermal, space, or heat exchangers. In the case of heat exchangers used in computers for processor heat dissipation, for instance, we can anticipate challenges in integrating dedicated flip-flop circuits and thermoelectric materials specifically designed for that purpose.

Finally, it is important to stress that we are not claiming that TeGraM is intended to replace, or even to be compared to, established memory technologies. We are proposing a new venue of development that could lead to niche applications. Interestingly, as heat and electronic flow have different regimes, some of these applications could even blur the current distinction between volatile and nonvolatile memories.

## 6. Conclusions

We put forward TeGraM, a temperature gradient memory that can be integrated into current electronic circuits. A proof of concept was demonstrated with a read/write-like process and hysteresis loop measurement. The proposal of TeGraM is a manifestation of the role hysteresis loop as a tool for addressing the capabilities of memory technology. Furthermore, this work solidifies hysteresis as a fundamental phenomenon connected to the thermodynamic irreversibility of computational memory processes.

## 7. Patents

The TeGraM described in this work was submitted for a patent application under process number BR 10 2024 011185 0.

## Figures and Tables

**Figure 1 entropy-27-00129-f001:**
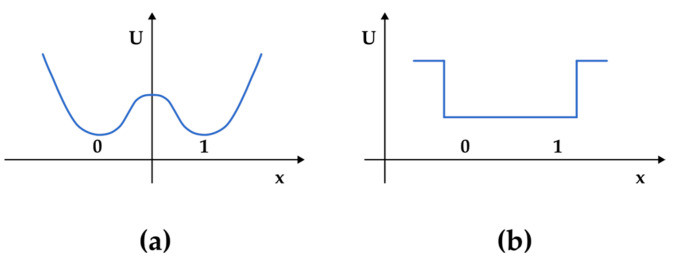
Two kinds of potential used to model nonvolatile memory devices according to Landauer. (**a**) The bistable potential well where x represents a generalized coordinate representing the parameter that is switched and U represents potential energy. (**b**) The potential well with 0 and 1 not separated by a barrier. In this case, the recorded data are maintained because random motion is slow.

**Figure 2 entropy-27-00129-f002:**
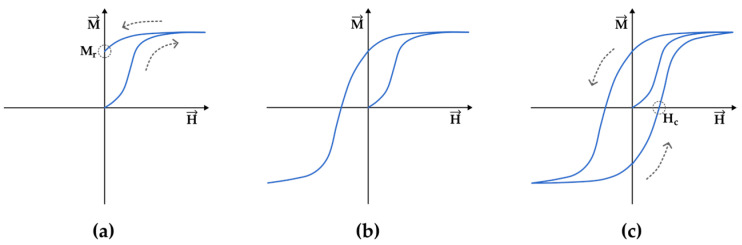
Sketched hysteresis curve illustrating the process of loop acquisition. (**a**) The ferromagnetic sample departs from the demagnetized state and has its initial magnetization curve up to saturation as the applied field (H) goes from zero to maximum. As H decreases back to zero, the irreversibility becomes evident as the magnetization does not follow the same path taken initially and the sample features a remnant magnetization M_r_. (**b**) The sample is magnetized up to saturation in the opposite direction. (**c**) The loop is closed as the sample is magnetized in the positive direction again.

**Figure 3 entropy-27-00129-f003:**
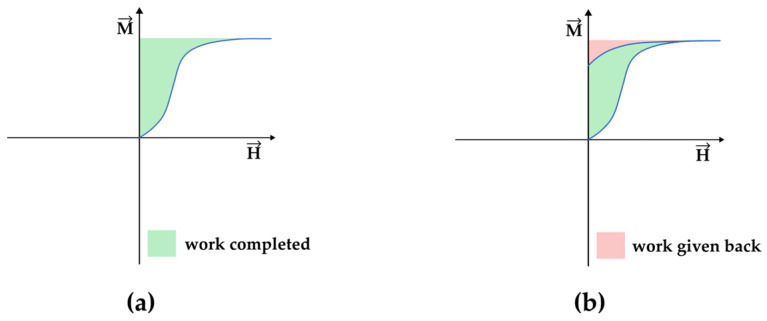
Evaluation of the work completed by the system according to Equation (1) for the hysteresis curve. It is verifiable that the internal area of the hysteresis loop is the energy dissipated by the volume of material. (**a**) The green shaded area represents the work completed during magnetization. (**b**) When the applied field returns to zero, part of the work, highlighted in red, is given back.

**Figure 4 entropy-27-00129-f004:**
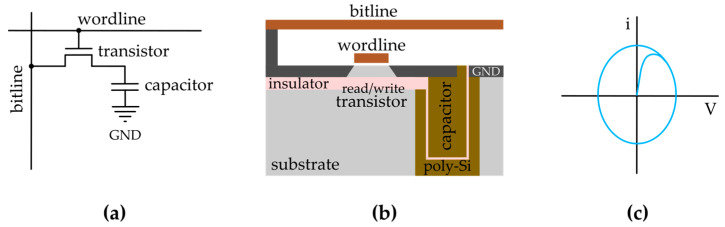
(**a**) Symbolic sketch representing a unit cell of DRAM. (**b**) Simplified layout for the realization of a trench capacitor DRAM unit cell in the semiconductor industry. (**c**) Sketch of a typical current–voltage curve for an RC circuit subjected to a sinusoidal voltage supply. Note how the curve starts from the uncharged state before establishing the characteristic elliptical shape. The resulting curve is hysteretic and represents both the information storage ability of a capacitor and the energy dissipated for each DRAM cell refresh cycle.

**Figure 5 entropy-27-00129-f005:**
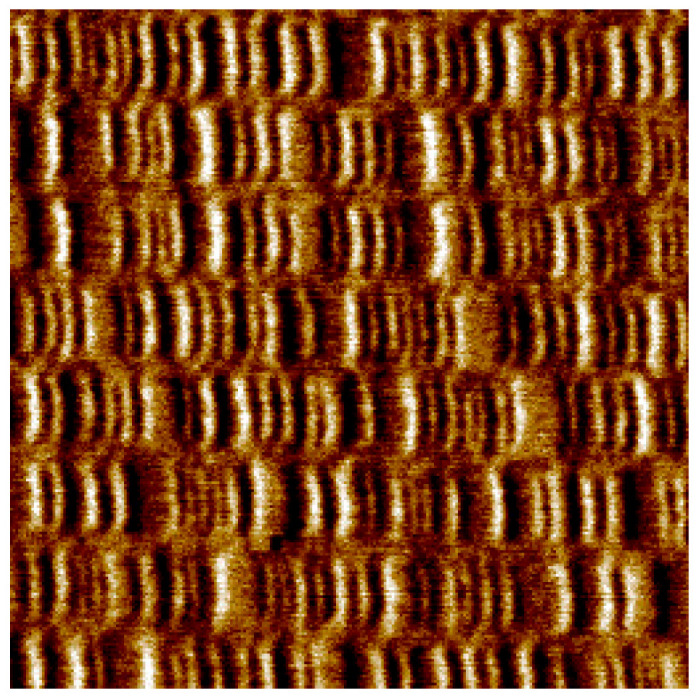
Magnetic force microscope (MFM) image of a 7 × 7 µm^2^ area of the surface of a disabled HDD. The binary information is stored as regions magnetized in opposite directions. As the read head flies over the track, it searches for magnetization reversal at a fixed frequency. If a reversal is detected, the information “1” is assigned. On the contrary, if no reversal is measured, the information “0” is assigned. In the image, a small stretch of 8 tracks is visible. Image obtained by the author JS.

**Figure 6 entropy-27-00129-f006:**
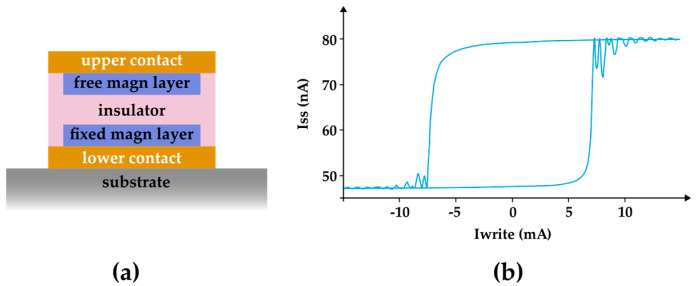
(**a**) The cell structure of an MRAM. (**b**) Hysteresis loop of a field-induced magnetic switching of an MTJ that is present in MRAM and STT-MRAM. This loop was sketched based on [16] and represents the relation between parameters related to the magnetic behavior of the free layer, where Iss is the current that crosses the junction and Iwrite is the current applied to the upper and lower contacts.

**Figure 7 entropy-27-00129-f007:**
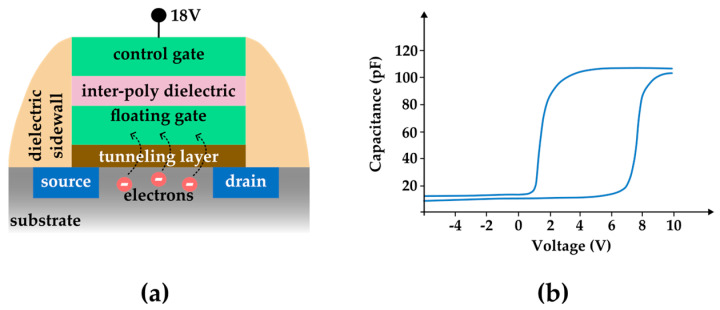
(**a**) Sketch of a flash memory cell illustrating the Fowler–Nordheim mechanism. A voltage is applied to the Control Gate. As a consequence, electrons are trapped in the floating gate. (**b**) Memory window measurement of a flash memory system consisting of Al/SiO_2_/TiBb_2_O_7_/SiO_2_/Si annealed at 900 °C. The memory window is 6.06 V in this case. This loop was sketched based on [19].

**Figure 8 entropy-27-00129-f008:**
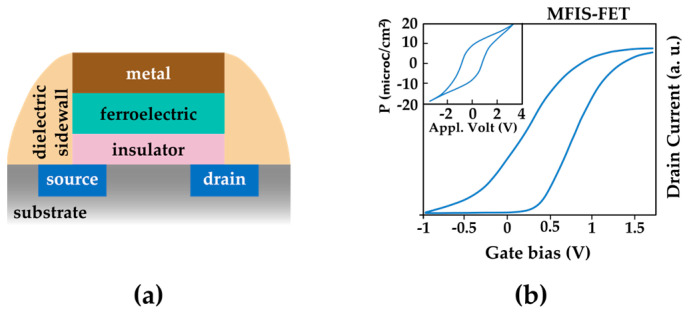
(**a**) Sketch of the cell structure of a Metal–Ferroelectric–Insulator–Semiconductor FET. (**b**) Memory window of a specific MFIS-FET resulting from the ferroelectricity of the Si:HfO_2_ layer highlighted in the upper inset. Loop sketched based on [23].

**Figure 9 entropy-27-00129-f009:**
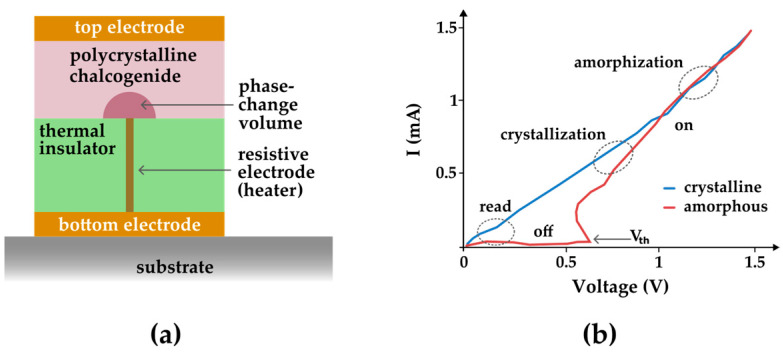
(**a**) The cell structure of PCRAM. (**b**) Hysteresis loop of a current-induced phase transformation that is present in PCRAM. This loop was sketched based on [17].

**Figure 10 entropy-27-00129-f010:**
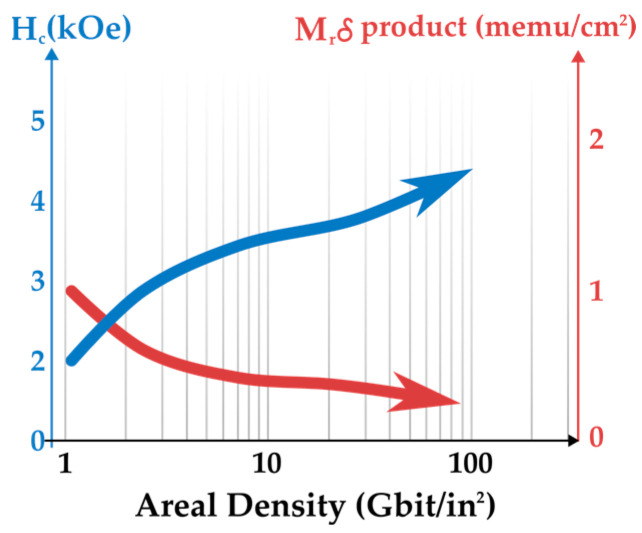
Evolution of the coercive field (H_c_) and media magnetic flux (M_r_δ) as a function of the increase in the aerial density of stored data in HDDs. We see that the coercive field of the media increases over time, whereas the magnetic flux shows the opposite tendency. The trend was sketched using data from [8].

**Figure 11 entropy-27-00129-f011:**
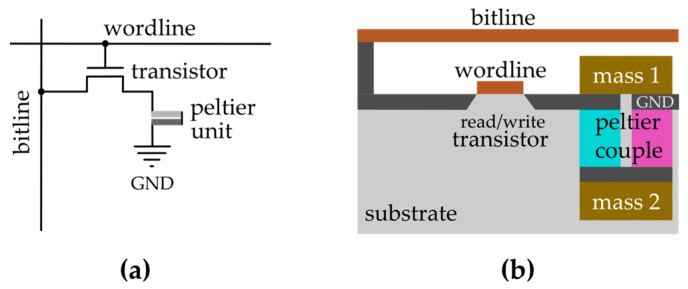
(**a**) Symbolic sketch representing a unit cell of aTeGraM. (**b**) Simplified layout for a realization of an aTeGraM unit cell in the semiconductor industry.

**Figure 12 entropy-27-00129-f012:**
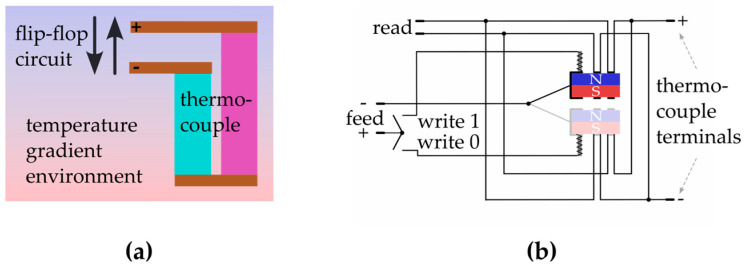
(**a**) Simplified layout for a realization of a pTeGraM unit cell. (**b**) Sketch of a possible implementation of a flip-flop circuit via a transruptor. A switch is used to write 1 or 0 by acting on a solenoid that pushes a magnetic connector between two distinct positions. Depending on the position of the connector, the constant input signal from the thermocouple can be inverted in the read terminal. Other flip-flop circuits can be implemented.

**Figure 13 entropy-27-00129-f013:**
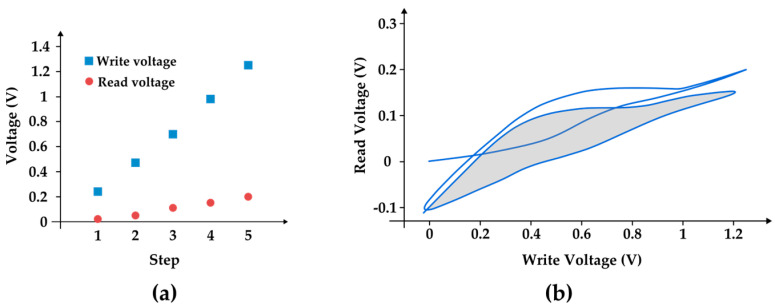
(**a**) Correlation between write and read voltages in an aTeGraM data cell. (**b**) Hysteresis loops correlating read and write voltages in the same aTeGraM data cell.

## Data Availability

The raw data supporting the conclusions of this article will be made available by the authors on request.

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
