# Peer review of "Temperature Gradients as a Data Storage Principle"

_entropy, 2025, doi:10.3390/e27020129_

Round 1
Reviewer 1 Report
Comments and Suggestions for Authors
The paper is devoted to the actual and important problem. It is well-written and clearly organized. It may be publishable after the appropriate revision.
Remarks:
1. In the context of the paper, it is noteworthy that the Landauer limit is referred today as the Carnot-Landauer limit, see:
Taranto Ph. et al., Landauer Versus Nernst: What is the True Cost of Cooling a Quantum System, PRX Quantum 4, 010332, 2023.
2. The Carnot engine has been addressed within the framework, of the Landauer principle based approach to the Second Law of Thermodynamics, see:
Ladyman J., et al. The use of the information-theoretic entropy in thermodynamics, Studies in History and Philosophy of Science Part B: Studies in History and Philosophy of Modern Physics, 39 (2), 2008, pp. 315-324.
3. The asymmetry of the writing and erasure of information in its relation to the Landauer principle should be calrified in the paper; the Landauer Principle holds for the erasure of information only.
4. The dimension of time was completely neglected by the author. The velocity of computation should be addressed by the authors in the context of the Landauer Principle, see:
Bormashenko Ed. Landauer Bound in the Context of Minimal Physical Principles: Meaning, Experimental Verification, Controversies and Perspectives, Entropy, 2024, 26(5), 423
Reviewer 2 Report
Comments and Suggestions for Authors
Summary of the Works
This research explores the connection between energy dissipation, hysteresis in physical systems, and data storage. It examines the thermodynamics of modern storage technologies, including RAM, HDD, MRAM, and others, in the context of Landauer’s principle and hysteresis. The study introduces the concept of Temperature Gradient Memory (TeGraM) and presents a practical device and experimental results. Emphasizing the second law of thermodynamics, the authors link hysteresis to irreversibility and energy loss, showing that this dissipation corresponds to the hysteresis loop's energy loss during switching.
Key findings
The authors:
i) Established that the feasibility of data storage technologies is intrinsically linked to their hysteresis loops.
ii) Leveraged this relationship to propose a novel data storage principle, Temperature Gradient Memory (TeGraM), inspired by heat engine concepts and compatible with modern electronic circuits.
iii) Showed a proof-of-concept through a read-write process and measured the associated hysteresis loop.
General Comments
- English should be double-checked; several typos were found, including in the abstract (e.g., “the Landauer’s principle” should be “Landauer’s principle”, etc.).
Positive aspects of this study
- The study links the hysteresis behavior of physical systems with data storage efficiency, offering a thermodynamic framework for understanding and optimizing modern storage technologies.
- The introduction of Temperature Gradient Memory (TeGraM) may represent a conceptual advancement, drawing inspiration from heat engines to propose a novel data storage paradigm that is compatible with existing electronic circuits. The proof-of-concept demonstration with a read-write process and hysteresis loop measurement provides strong experimental support for the theoretical foundations, enhancing the credibility of the proposed TeGraM technology.
- The positive aspect is that by integrating principles from thermodynamics, physics, and electronics, the research bridges multiple disciplines, paving the way for further exploration of energy-efficient and sustainable data storage solutions.
Vulnerable points
- However, relying heavily on analogies, such as those drawn from thermodynamics, can indeed lead to skepticism, particularly if the analogies are not accompanied by rigorous derivations, extensive experimental validation, or clear connections to established principles in the field of data storage. The work may risk being seen as speculative or lacking practical relevance without further grounding.
- Collaborations with experts in materials science, electronics, and computational modeling are crucial for providing additional credibility. Incorporating interdisciplinary perspectives would help ensure the theoretical foundations are robust and aligned with practical implementations.
Specific Questions
1) How does the hysteresis loop area quantitatively affect the energy efficiency and thermal management of various data storage technologies?
2) What specific thermodynamic principles of heat engines were adapted to develop the Temperature Gradient Memory (TeGraM) concept, and how do they enhance its integration into electronic circuits?
3) Figure 6(b) illustrates the hysteresis loop corresponding to the field-induced magnetic switching of a Magnetic Tunnel Junction (MTJ), a key component in MRAM and STT-MRAM. This loop depicts the relationship between parameters associated with the magnetic behavior of the free layer, where 𝐼𝑠𝑠represents the current passing through the junction, and 𝐼𝑤𝑟𝑖𝑡𝑒denotes the current applied to the upper and lower contacts. Could you explain the physical origin of the oscillations observed near the transition points at approximately −7 mA and +7 mA?
4) In the proof-of-concept experiment, how were the hysteresis loop measurements correlated with data read-write fidelity and performance under varying temperature gradients?
5) What are the limitations or challenges in scaling the TeGraM concept for high-density data storage systems, particularly in terms of energy dissipation and thermal stability?
6) How does the hysteresis-based model proposed in this research compare with existing theoretical models for data storage efficiency and irreversibility?
7) What potential advancements could further minimize the energy dissipation in data storage systems while maintaining reliable hysteresis loop characteristics?
8) Are there specific materials or device architectures identified as particularly suitable for implementing TeGraM, given its reliance on temperature gradients and hysteresis behavior?
9) We may object that the proof-of-concept presented, while promising, may not be sufficient to convince skeptics without comprehensive testing across different scenarios and scales. Demonstrating the feasibility of TeGraM in real-world applications and showing how it outperforms or complements current technologies would enhance its perceived rigor and applicability. Authors are asked to express their opinions on the matter
Conclusions
The work presents innovative aspects, notably the introduction of the TeGraM concept, but it also exhibits certain vulnerabilities that need to be addressed. To mitigate the concerns outlined earlier, the authors could enhance their research by incorporating validation through traditional, well-established methodologies in data storage. This could include comparing their approach with existing technologies, benchmarking energy efficiency, and evaluating performance metrics under realistic operational conditions.
Comments on the Quality of English LanguageEnglish should be double-checked; several typos were found, including in the abstract (e.g., “the Landauer’s principle” should be “Landauer’s principle”, etc.).
Round 2
Reviewer 2 Report
Comments and Suggestions for Authors
The authors have addressed the questions raised in my initial report point by point. Their data storage principle, "Temperature Gradient Memory (TeGraM)," which is inspired by heat engine concepts and designed to be compatible with modern electronic circuits, remains in its early stages. The validation of this principle requires further investigation and development, a step that is crucial to establish the robustness of its theoretical foundations and its potential for practical implementation. Overall, I believe this version of the manuscript is suitable for publication.